# Changes in Chemical Composition and Fatty Acid Profile of Milk and Cheese and Sensory Profile of Milk via Supplementation of Goats’ Diet with Marine Algae

**DOI:** 10.3390/ani13132152

**Published:** 2023-06-29

**Authors:** Ferenc Pajor, Dávid Várkonyi, István Dalmadi, Klára Pásztorné-Huszár, István Egerszegi, Károly Penksza, Péter Póti, Ákos Bodnár

**Affiliations:** 1Department of Animal Husbandry Technology and Animal Welfare, Institute of Animal Sciences, Hungarian University of Agriculture and Life Sciences, Páter Károly 1, 2100 Gödöllő, Hungary; varkonyid99@gmail.com (D.V.); egerszegi.istvan@uni-mate.hu (I.E.); poti.peter@uni-mate.hu (P.P.); bodnar.akos@uni-mate.hu (Á.B.); 2Department of Livestock Products and Food Preservation Technology, Institute of Food Science and Technology, Hungarian University of Agriculture and Life Sciences, Ménesi út 43-45, 1118 Budapest, Hungary; dalmadi.istvan@uni-mate.hu (I.D.); pasztorne.huszar.klara@uni-mate.hu (K.P.-H.); 3Department of Botany, Institute of Agronomy, Hungarian University of Agriculture and Life Sciences, Páter Károly 1, 2100 Gödöllő, Hungary; penksza.karoly@uni-mate.hu

**Keywords:** fatty acids, cheese, e-nose, DHA, *Schizochytrium limacinum*

## Abstract

**Simple Summary:**

Recently, global interest in goat milk and goat milk products has increased, due to their high nutritional value and better amino acid composition and fatty acid profile compared to cow milk. Moreover, the more favorable fatty acid composition of these foodstuffs (such as increased long-chain n-3 fatty acids) has significant implications on the reduction of human health problems and is increasingly important for consumers as a result. One of the common ways to enhance the foodstuffs via long-chain n-3 polyunsaturated fatty acids (PUFA) is to enrich animal diets with fishmeal, fish oil, or marine algae. Marine algae are a source of long-chain n-3 PUFA, such as docosahexaenoic acid (DHA). DHA is essential for human nutrition because it is able to reduce the risk of coronary heart disease and it has antihaemolytic properties. Conversely, marine algae supplementation is linked to some adverse effects, such as decreased dry matter intake and milk yield, and results in milk fat depression in dairy animals and influences the flavour of their milk. This study evaluated low-level marine algae supplementation in goat milk, whey, and cheese composition, and the fatty acid profile of goat milk and cheese as well as the sensory profile of goat milk. In this report, daily supplementation of 5 g/head of marine algae significantly influenced the recovery of fat and protein in curd, whey, and cheese composition. In addition, marine algae supplementation markedly improved the concentrations of beneficial fatty acids in milk and cheese without negative effects on milk flavour.

**Abstract:**

The aim of the present study was to assess the effects of the low level of *Schizochytrium limacinum* marine algae (daily 5 g per animal) on the milk, cheese, and whey composition; fatty acid profile of milk and cheese; and the sensory profile of goat milk using an e-nose device. Thirty Alpine goats were randomly divided into two groups: the control group (C, *n* = 15)—fed grass with daily 600 g concentrate and the experimental group (MA, *n* = 15) who received the same forage and concentrate supplemented with 5 g/head/day marine algae. Animals were kept indoors and the investigation period lasted 52 days, including the first six weeks as the period of adaptation and the last 10 days as the treatment period. During the adaptation period, bulk milk samples from each group were collected once a week (0, 7, 14, 21, 28, 35, and 42 d), while during the treatment period (10 days), bulk milk samples from each group were taken every day, and cheese samples were processed from bulk milk each day from both groups. Marine algae supplementation had no negative effect on milk composition. In contrast, the marine algae inclusion significantly elevated the fat and protein content of whey and the protein content of cheese, as well as the recovery of fat and protein in the curd, while increasing the cheeses’ moisture content on a fat-free basis. The marine algae supplementation significantly increased the docosahexaenoic acid (DHA) and the rumenic acid (CLA c9t11) concentrations and decreased the n-6/n-3 ratio in the milk and cheese. There were no significant differences between the C and the MA group with regard to the sensory profiles of the milk. It can be concluded that the milk obtained from goats given daily supplementation of 5g of MA has a fatty acid profile more beneficial to human health, without any negative effects on the milk’s aromatic components.

## 1. Introduction

Global goat milk and cheese production reached approximately 21 million tons in 2021 and 511 thousand tons in 2020 [1]. The demand for goat milk and goat milk products (e.g., cheese, yoghurt, and butter) has increased over the last decades due to their high nutritional value (such as amino acid composition) and higher digestibility (smaller fat globules and softer curd formation of its proteins) compared to cow milk and cow milk products [2].

The production of goat milk and its products could play a significant role in contributing to retaining people in rural areas and creating jobs, due to the consumer demand for premium-quality goat dairy products [3].

Nowadays, there is a markedly increased interest in modifying the fatty acid composition of milk and dairy products. Researchers have generally focused on high-content n-3 polyunsaturated fatty acids (PUFA) feed supplements (such as oils, seeds, or freshwater and marine algae) [4,5,6]. One of the most interesting n-3 PUFAs is docosahexaenoic acid (DHA). DHA has beneficial effects on human health, such as reducing the risk of coronary heart disease [7], as well as antihaemolytic properties [8]. Moreover, DHA is essential for the functional development of the brain in infants and normal brain function in adults; in addition, DHA in the human diet improves learning ability [9]. Nevertheless, only fish and marine algae products (e.g., oil, powder) contain notable DHA concentrations [10]. The importance of DHA in human health has been well proven by the European Food Safety Authority (EFSA) Panel on Dietetic Products, Nutrition, and Allergies is recommended for human consumption. Presently, the recommended daily intake of eicosapentaenoic acid (EPA) + docosahexaenoic acid (DHA) fatty acids for the adult population is 250 mg, while adequate daily intake for the children is 100 mg DHA [11].

The different concentrations of marine algae supplementation (e.g., *Schizochytrium limacinum*) in dairy animals’ diets (regular doses of level of marine algae supplementation: 8 to 36 g/kg DM intake) have significantly increased DHA concentration in milk [12,13,14,15]. In line with this, some studies have reported that a ruminant diet enriched with marine algae (level of MA from 8 to 36 g/kg DM intake) resulted in decreased dry matter intake, milk yield, and milk fat depression in dairy animals [13,14,15], and decreased the population of protozoa and cellulolytic bacteria in their rumen [10,12,16]. Moreover, the impact of *S. limacinum* supplementation on the products’ sensory profile needs to be evaluated as this may affect the acceptance of milk by consumers. The flavor (e.g., odour, aroma) is an important parameter for consumer acceptance of milk and milk-based products, especially goat milk products. The sensory evaluation of dairy products commonly incorporates a sensory acceptance test conducted by a panel of consumers who are trained according to the relevant ISO standard [17]. The use of artificial sensors is a new trend among modern analytical techniques. The results of using these sensors are very similar to those given by panellists. The use of an electronic nose (e-nose) and electronic-tongue (e-tongue) are commonly and successfully applied analytical methods [18].

To the best of our knowledge, there are no existing studies in the literature on the effects of the dietary supplementation with *Schizochytrium limacinum* marine algae on goat cheese and whey parameters (yield, composition, and recovery of fat and protein in the curd), and the sensory profile of goat milk.

We hypothesized that the inclusion of a low concentration of marine algae (daily 5 g per animal) in the diet of goats significantly influences the fatty acid profile (such as DHA and rumenic acids) of their milk and cheese, without significant effects on milk fat depression and cheese-making parameters or the sensory profile of their milk.

The present study aimed to assess the effects of supplementing marine algae, at a rate of 5 g/head/day, on the composition of goat milk, cheese, and whey; the fatty acid profile of goat milk and cheese; and the sensory profile of goat milk.

## 2. Materials and Methods

### 2.1. Experimental Design

The experimental procedures and animal care conditions were in compliance with the European guidelines for the care and use of animals in research (Directive 2010/63/EU). The study was carried out in an Alpine goat farm (Csitár, Nógrád County, Hungary; geographical coordinates: 48°02′54.2″ N 19°25′11.8″ E).

A total of 30 milking Alpine does were involved in this study, which were balanced for time of kidding (early March), kid rearing (8 weeks), parity (2nd parity), and udder health parameters (no detected clinical mastitis symptoms, as swelling, heat, redness, or pain). After weaning, all goats were milked twice a day by machine milking. All goats between morning (06:00) and evening (18:00) milking continuously grazed on the natural pasture and overnight, they were kept in a barn.

The investigation period lasted 52 days, including the first six weeks as the period of adaptation and the last 10 days as the experimental period. Thirty goats (days in milk (DIM) 34 ± 2.56 d) were randomly allocated to two groups (milk yield in control group: 1.28 kg/d; milk yield in experimental group: 1.24 kg/d, respectively).

The animals in the control group (C, *n* = 15) were fed grass and 600 g commercial concentrate (17% crude protein; 7.77 MJ/kg Net Energy of lactation (NEl)); in the second group (MA, *n* = 15), goats received the same feed supplemented daily with 5 g/head of dried *Schizochytrium limacinum* marine algae and mixed manually into the daily concentrate portion for each goat. The control and the experimental concentrates were isonitrogenous and balanced by net energy content. In both groups, the concentrate was individually offered to the all goats twice a day in identical amounts, before milking, without refusals. The diets were adjusted to the National Research Council recommendations of energy and protein requirements for dairy goats [19]. A commercial trace-mineralized salt block and drinking water were provided ad libitum to all animals.

The dried marine algae supplement was produced by Alltech Inc. (product name: ALL-G-RICH^®^; Dunboyne, Co Meath, Ireland) (chemical composition of marine algae: dry matter (DM): 929 g/1000 g, crude protein: 148 g/kg DM, crude fat: 482 g/kg DM, crude fiber: 23 g/kg DM, ash: 38 g/kg DM).

The natural pasture was grazed on extensively by the animals during the investigation period; two groups of goats grazed together on a pasture, where no artificial fertilizers were used during pasture management. Stocking density was less than 0.5 animal unit (AU)/ha, in order to avoid over-grazing of the area. Total biomass yield of pasture was calculated from the difference between the grass yield before and after the grazing period. A day before grazing, five 2 × 2 m plots were harvested by hand clipping at a height of 3 cm above the ground, and fresh weight was weighed using a scale. After grazing, a similar procedure was carried out to determine the grass yield [20].

The annual grass yield (green) was 1.7 t/ha green yield. The main grass species were *Festuca pseudovina*, *Lolium perenne, Festuca arundinacea*, and *Digitaria sanguinalis*; the main legumes were *Lotus corniculatus* and *Medicago lupulina*. The mean annual temperature was 9.0 °C, and the total annual precipitation was 610 mm in the pasture area.

The composition of the experimental diets is shown in Table 1.

### 2.2. Collection of Milk Samples

All goats were milked twice a day during the investigation period (52 d). Daily bulk milk yield was recorded for each group, then the daily milk yield (DMY) was calculated as follows: daily bulk milk yield per 15 animals. Bulk milk samples from each group were collected once a week during adaption period (0, 7, 14, 21, 28, 35, and 42 d), and every day during experimental days (10 days) into 2 × 50 mL plastic tubes for chemical composition, and fatty acid and sensory analysis. Milk samples were stored at 4 °C for the later analysis, except samples for fatty acid analysis, which were frozen and stored at −80 °C before laboratory analysis.

### 2.3. Cheese Processing and Collection of Cheese Samples

Cheese samples (n = 10) were prepared from bulk milk every day independently during the investigation period from both groups. On each cheese processing day, goat cheeses (400~ g) were made from goat bulk milk. Remaining whey was measured each day. Cheese samples were collected after processing, frozen, and stored at −20 °C until further analysis. The description of cheese processing is shown in Table 2.

Cheese yield was expressed as kg per 100 kg of goat milk used. Moisture-adjusted cheese yield on a 440 g/kg total solids basis was calculated by [21] as follows:Moisture-adjusted cheese yield = (total solids of milk − total solids of whey)/(440 − total solids of whey) × 100.(1)

Moisture on a fat-free basis (MFFB, %) was calculated as follows:MFFB = (100 − total solids)/(100 − fat content) × 100.(2)

Percentage of fat recovery (3) and protein recovery (4) in the curd for each batch was calculated as [22]: Recovery of fat = (% fat in cheese × cheese wt)/(% fat in milk × milk wt)(3)
Recovery of protein = (% protein in cheese × cheese wt)/(% protein in milk × milk wt)(4)

Cheese fat in total solids (TS) was calculated as follows: fat/TS × 100. Moreover, whey parameters (fat, protein and lactose) in total solids were calculated as follows: fat/TS × 100, protein/TS × 100 and lactose/TS × 100.

### 2.4. Chemical Analysis

Samples from grass, concentrate, and marine algae were taken at the start of the trial and were analyzed for dry matter, crude protein, crude fat, crude fiber, and crude ash, according to the procedure of the Hungarian Feed Codex [23].

The morning and evening daily bulk milk samples were mixed before analysis. Fat, protein, lactose, and total solid content of milk and whey were determined using the LactoScope™ IR spectrometry analyzer (Delta Instruments, Drachten, The Netherlands).

The dry matter, fat and protein content of cheese samples were analysed as described in the Hungarian Standards [24,25,26]. (Hungarian Standard, 1978, 1980, and 2002).

The grass, concentrate, and marine algae meal, as well as milk and cheese fat were extracted with the method developed by Gerber [27]. Fatty acids (FA) were re-esterified to methyl esters using the procedures according to ISO 12966-2 (2011) standard [28]. Methyl esters of fatty acids (FAMEs) were determined via gas chromatography (gas chromatographer GC 2010, Shimadzu, Kyoto, Japan) with a flame ionization detector (FID) and a column (Zebron ZB-WAX, 30 m × 0.25 mm × 0.25 μm). The split injection ratio was 50:1. Helium was used as the carrier gas, applying a flow rate of 28 cm/s. The injector and detector temperatures were 270 and 300 °C, respectively. The oven temperature programmed run started at 80 °C, then was increased 2.5 °C/min up to 205 °C and held for 5 min, and then increased again to 250 °C at 10 °C/min and was held for 5 min at 210 °C. Peaks were identified on the basis of the retention times of standard methyl esters of individual FAs (Supelco 37 Component FAME Mix, Sigma-Aldrich, St. Louis, MO, USA). The individual FAs were calculated using the ratio of their peak area to the total area of all observed acids. In this study the data obtained were expressed as % of total FAMEs. The selected FA combinations were calculated by using FA data: saturated fatty acids (SFAs); monounsaturated fatty acids (MUFAs); polyunsaturated fatty acids (PUFAs); total n-6 and n-3 PUFA and n-6/n-3 ratio. Atherogenic index (AI) was calculated according to Ulbricht and Southgate [29].

### 2.5. Sensory Analysis of Milk

Head space analysis of samples was performed using an NST3320-type electronic nose (Applied Sensor, A.G., Sweden) with a built-in headspace autosampler unit for 12 samples. The sample chamber contains 23 different sensors, and software for collecting and processing the data of the specimen. The NST 3320 is equipped with 10 MOS-FET (metal oxide semiconductor field effect transistor) sensors, 12 MOS (metal oxide semiconductor) sensors, and a sensor for relative humidity measurements. Ambient air was used as a reference gas for the sensors, which was filtered through a silica gel column and a combined moisture/hydrocarbon filter. The gas-flow rate of the dynamic sampling was set to 50 mL/min. The milk samples were kept at 10 °C until starting the measurements. 6 mL of milk samples was measured into 30 mL vials used for gas chromatography measurements, and then sealed with Teflon-coated closure elements. The electronic nose measurement sequence started with sample equilibration at 25 °C for 20 min. Then, reference air was pumped over the sensor surfaces for 10 s (baseline) followed by the infusion head-space for 30 s (sampling time) while the sensor signals were recorded. After sample analysis, the recovery phase of the sensors was set to 260 s including the flush time of the gas lines (60 s) with filtered air prior to the next sample injection to allow the re-establishment of the instrument baseline. The total cycle time per sample was 300 s. Each sample was measured 5 times and the results of 10 measurements from 2 replicate samples were used for the statistical analysis.

### 2.6. Statistical Analysis

Statistical analysis was performed using the SPSS 27.0 software package. Shapiro–Wilk’s test was used for testing the normality distribution. The effect of diet (C and MA), on milk, whey, and cheese composition, and the fatty acid composition of milk and cheese samples were determined via analysis of variance (ANOVA). Differences are shown when *p* < 0.05.

Multivariate statistical methods, namely principal component analysis (PCA) and canonical discriminant analysis (CDA) were used for the evaluation of electronic nose results. Principal component analysis is a commonly used method for separating groups in the high-dimension data space. As a nonparametric test, the separation is based on the inner correlation of data, so it can be tested whether well separated groups can be formed in a natural way. Canonical discriminant analysis is one of the most frequently used of parametric classification procedures. The method maximizes the variance between categories and minimizes the variance within categories. The discriminant functions are the linear combinations of the standardised independent variables (responses of electronic nose), which yield the biggest mean differences between the groups.

## 3. Results

### 3.1. Milk, Cheese and Whey Composition

The average daily milk yield and milk composition are reported in Table 3. At pre-treatment, daily milk yield and milk parameters were consistent between the two groups. The marine algae supplementation did not affect the daily milk yield of goats and chemical composition of milk. The mean value of the milk fat and protein content were 3.59% and 3.22% for the experimental group (MA) and 3.51% and 3.24% for the control group (C).

The cheese yields and composition are shown in Table 4. The actual cheese yield, fat content, and fat content in total solids were similar between the two groups. In contrast, the marine algae supplementation did considerably affect the moisture-adjusted cheese yield, MFFB value, protein content, and total solids. The mean value of the moisture-adjusted cheese yield, MFFB value, protein content and total solids were 14.32 kg/100 kg of milk, 69.4%, 20.08%, and 46.10% for the control group (C) and 13.0 kg/100kg of milk, 72.28%, 17.83%, and 44.10% for the experimental group (MA), respectively. MA supplementation was strongly influenced the recovery of fat and protein in the curd, and significantly decreased them in the MA group compared to the C group (fat: 72.4% vs. 79.6%; protein: 66.6% vs. 77.7%).

The whey composition is shown in Table 5. During the experiment, the lactose and total solids content were consistent between the two groups. In contrast, the marine algae supplementation had a significant effect on the whey fat and protein content. The whey fat and protein content were significantly higher in the MA group compared to C (1.04 and 1.20% vs. 0.60 and 0.85%). Moreover, the fat, protein and lactose content in the total solids also differed by diet.

### 3.2. Milk and Cheese Fatty Acid Composition

The fatty acid profile in milk and cheese samples is presented in Table 6.

The daily 5 g per animal marine algae supplementation had a substantial impact on the milk’s fatty acid profile. The concentrations of capric acid (C10:0), myristic acid (C14:0), myristoleic acid (C14:1), palmitic acid (C16:0), palmitoleic acid (C16:1) (only in milk samples), rumenic acid (c9t11 C18:2), docosahexaenoic acid (C22:6), saturated fatty acids, and AI were significantly increased, while the concentrations of stearic acid (18:0), oleic acid (c11 C18:1), linoleic acid (C18:2), alpha-linolenic acid (C18:3), docosapentaenoic acid (C22:5) (only in milk samples), monounsaturated fatty acids, n-6 fatty acids, and n-6/n-3 ratio were markedly decreased in the milk and cheese fat of experimental animals.

The calculated daily DHA intake was 673.91 mg on the whole duration of the treatment (Table 7). The average milk DHA content and DHA conversion efficiency ratio resulting from the marine-algae-enriched diet to milk was continuously increased during the treatment and reached the highest values at the sixth week. And these values remained consistent until the end of the treatment.

### 3.3. Sensory Profile of Milk

Signal responses of the electronic nose were first processed via principal component analysis. We used a multivariate statistical method to investigate whether the data obtained from the 23 sensors of the electronic nose showed any distinctly separable group formation as a result of supplementary feeding. Figure 1. shows the sample points in the score plot of the first two principal components. The two principal components together account for 68% of the total variance. The position of the measurement points shows that the daily 5 g per animal marine algae supplementation did not cause such a significant change in the volatile component profile of the samples analysed via principal component analysis.

The original electronic nose data were also analysed also via canonical discriminant analysis. Since the method uses information about which group the given sample was classified into in the model creation process, in most cases it can distinguish between groups more effectively than PCA. The confusion matrix shows that 95% of the repeated measurements were successfully re-classified into the correct group based on the created model (Table 8). However, the success rate was reduced by the results of cross-validation. It can be observed that, in this case, the model was able to make a correct determination of which group a sample belonged to in only 45% of the cases, based on the given measurement results. The control sample was classified into the correct group only 40% of the time, and 50% of milk samples of goats fed daily with 5 g marine algae supplementation per animal were correctly classified.

## 4. Discussion

### 4.1. Milk, Cheese, and Whey Composition

The values of the investigated milk parameters were comparable with earlier reports, and these values were within the normal ranges for dairy goats [30,31,32]. The milk constituents were not affected by marine algae supplementation due to the low amounts of daily marine algae supplementation. Our results were consistent with previous reports, where the fed diets contained low concentrations of marine algae, e.g., as daily 105 g/head/day for dairy cows [33], and 10g/head/day for dairy goats [32].

Nevertheless, the milk composition of the dairy animals was influenced by the higher dose of marine algae. Previously, some authors have reported [13,14,15] that marine algae supplementation (level of MA from 8 to 36 g/kg DM intake) markedly reduced the milk fat content in cow, ewe, and goat milk. In contrast, other authors [32,34] have reported that the fat and protein content of milk were increased when diets were supplemented with marine algae (applied marine algae concentrations were 23.5–94 g/head/day, and 10 g/head/day). Nevertheless, milk components such as protein and fat content are important during milk-processing, as these have a great effect on cheese chemical composition and yield.

The actual cheese yield for the control group was 12.74 kg/100 kg of milk and for experimental group, it was 12.03 kg/100 of milk. The actual cheese yields obtained in this study were similar in comparison to soft cheeses [22,35]. Thus, these authors reported that the average yield of soft cheese made from goat or cow milk was about 12-13 kg/100 kg. While the moisture-adjusted cheese yield for the control group was 14.32 kg/100 kg of milk, and for experimental group, it was only 13.0 kg/100 of milk, as expected because of its higher moisture content. Furthermore, according to the European Commission, the cheese samples could be classified as a soft cheese, based on their moisture content on a fat-free basis (MFFB, <68%) [36]. There were no significant differences between the C and MA groups in terms of the fat content and total solids of the cheese. In contrast, in the experimental groups, the cheeses’ protein contents were lower compared to the control group. Regarding the chemical composition of the cheese samples, the results obtained were very similar to the data reported in [37,38]. 

In our study, the marine algae supplementation caused not only lower cheese protein contents, but also led to lower recovery ratios in the curd in the MA group compared to the C group. Kalit et al. [22] reported that the recovery of fat and protein were 80.1% and 74.4% in cattle, while others reported that the recovery of fat was 75–79%, and the recovery of protein was 80–81% in goats [39,40]. To the best of our knowledge, these recovery parameters have not been previously evaluated in milk from marine-algae-supplemented goats. Some authors have reported that oilseed (such as flaxseed, olive seed, soybean) supplementation had no effect on cheese yield and cheese chemical composition in dairy sheep [41] and cows [42]. Similar findings have also been reported by Zang et al. [43].

There were no significant differences between the C and the MA group with regard to the lactose and total solids content of whey. In contrast, the fat and the protein content of the whey in the experimental group were significantly higher compared to the control group due to the different recovery ratios of fat and protein in the curd. In the MA group, the recovery rate of protein and fat in the curd were markedly lower than in the C group. Similar goat-whey composition parameters have been reported for a control group [44]. In earlier reports, goat whey was reported to contain approximately 7% (wt/vol) dry matter, 4.5% (wt/vol) lactose, 0.7% (wt/vol) fat, 0.8% (wt/vol) protein, and 1% (wt/vol) salts [44,45,46]. Heino et al. [45] found only a small difference in whey protein content, depending on the type of cheese and cheese processing technology used, although whey fat content is strongly influenced by the curd-cutting process during goat cheese production [45,46].

### 4.2. Milk and Cheese Fatty acid Composition

The marine algae feeding did not influence the concentrations of short-chain fatty acids, such as butyric (C4:0), caproic acid (C6:0), and caprylic acid (C8:0). In contrast, in the MA group, the capric acid content (C10:0) was significantly higher that the C group. An earlier study [47] reported that short-chain fatty acids have a marked effect on the aromatic compounds of dairy products. The organoleptic properties of milk are important for consumers, since significant differences in these parameters were not found between the MA and the C group. In this study, the effects of MA supplementation on the sensory profile of milk was also tested. Recently, short-chain saturated fatty acids are becoming of greater interest to nutritionists. The consumption of these fatty acids can help in the prevention of malabsorption syndrome, due to rapid hydrolyzation and direct absorption through the gut wall to the liver portal vein [48,49].

The marine-algae-enriched diet elevated the concentrations of lauric acid (C12:0), myristic acid (C14:0), and palmitic acid (C16:0), due to the high concentration of these fatty acids in the experimental diet. This is in accordance with previous results [13,32].

The odd-chain fatty acid (OCFAs) concentrations were similar in the MA and the C group. It is well known, that OCFAs are indicators of ruminal fermentation, because these FAs are mostly produced by the rumen’s bacterial populations [50]. The OCFAs are absorbed via the intestinal wall and taken up by the mammary gland from the blood.

The MA supplementation significantly decreased the concentrations of stearic acid (C18:0) and oleic acid (*c11* C18:1) in milk and cheese. Daily marine algae inclusions of 5 g/head in the goats’ diet had a significant impact on biohydrogenation in the rumen. Previous authors have reported [51] that the presence of long-chain PUFAs in the animals’ diet inhibits the PUFAs saturation to C18:0, leading to higher level of *trans* C18:1 (trans vaccenic acid—TVA) fatty acid and other specific biohydrogenation intermediates, which negatively influence the viability of protozoa and cellulolytic bacteria populations in rumen and which are associated with milk fat depression [10,12,13,14,15,16]. Therefore, the MA supplementation markedly decreased the amount of stearic acid and notably increased the TVA content in milk and cheese fat. TVA is a precursor of rumenic acid; TVA is converted to rumenic acid by Δ9-desaturase in the mammary gland [52]. It is well known that rumenic acid has beneficial effects on human health, such as suppressing carcinogenesis, modifying the immune system, and reducing atherogenesis [53,54]. Moreover, previous authors have reported that the marine algae enriched diet (g/kg DM basis) had a greater impact on milk fat rumenic acid (*cis*-9, *trans*-11) concentrations compared plant oil supplementation [55].

The reduced amount of stearic acid affected the amount of oleic acid (c11 C18:1) in milk and cheese. The concentration of oleic acid was significantly decreased in the MA group compared to the C group. It has previously been reported that 40% of the total amount of oleic acid in milk originated from the diet, whereas the remainder was synthesized in the mammary gland by stearoyl-CoA desaturase enzyme from stearic acid [56].

Concentration of DHA fatty acid in milk and cheese was significantly increased by the marine algae supplementation. DHA is required for many metabolic processes and has many positive effects on human health, such as reducing the risk of coronary heart disease [7]. In addition, DHA is essential for the normal operation of the brain, and improves memory and learning ability [9]. Unfortunately, the rate of alfa-Linolenic acid conversion to DHA is very low, and this ratio is below 0.6% in primates [57], and in humans (infants and adults) it is also lower than 1% [58]. Consequently, direct intake of DHA is more effective and sufficient DHA in the human diet is important. In our study, the mean DHA concentration in the MA group increased to 0.25 g/100g from 0.06 g/100 of fatty acids in milk, and similarly, to 0.24 g/100g from 0.05 g/100 of fatty acids in cheese; this is a more than 4-fold difference. Similar findings have also been reported by Toral et al. [59] and Pajor et al. [12]. They found that a marine algae enriched diet elevated DHA concentration in milk. Furthermore, similar findings have also been reported by Zisis et al. [60].

In this study, the AI was higher in the MA group compared to the C group. This index was increased, which is associated with a high level of palmitic acids and a low concentration of oleic acid, which negatively affected the concentration of MUFA in milk and cheese. This result suggests that the effect of the low level of MA supplementation on health benefits is uncertain. However, the MA supplementation in the diet demonstrated that it can increase the concentrations of beneficial fatty acids (rumenic acid, DHA) in milk and cheese, resulting in a more adequate fat for human nutrition.

In addition, the n-6/n-3 ratio was significantly decreased by marine algae supplementation both in milk and cheese. This is consistent with earlier reports [12,13]. The n-6/n-3 ratio is generally used to assess the nutritional value of animal-derived fats. Currently, the recommended value of the n-6/n-3 ratio by nutritionists is less than 4 [61]. The lower n-6/n-3 ratio in the milk and cheese of goats that received MA is in line with the new recommendations for human nutrition [11].

In the present study, the DHA transfer efficiency from algae to milk continuously increased and reached the highest value of 18.42% at 6 weeks (42 days), and remained relatively constant around this value during the experimental period. A similar result has previously been found [33]. The mean value of DHA transfer efficiency in the MA experimental period (between 43 and 52 days of treatment) was 17.42%. Other studies have reported that DHA transfer efficiency from an MA-enriched diet to milk fat was between 8 and 12% in different treatments [4,12,32]. Nevertheless, these values were higher than those in diets containing fish oil or meal; the DHA transfer efficiency ratio from fish-oil- or meal-supplemented diets to milk fat was less than 5% in dairy animals [62].

### 4.3. Sensory Profile of Milk

The applicability of chemical sensor arrays are becoming more and more widely demonstrated. Instrumental testing methods using non-specific sensors can support food research and product manufacturing in areas such as proof of origin, detection of adulteration, controlling freshness, monitoring storage stability, testing production stability, quality assurance, and product development [63]. For these purposes, intelligent sensory methods can also be successfully used with dairy products. Yang et al. [64] were able to distinguish goat milk samples from each other based on the different intensities of their flavour, with an efficiency of 98.2% and 100%, using an electronic nose with metal oxide sensors. In the present study, the control goats’ milk samples and the samples from goats given daily supplementation of 5 g of marine algae per animal were not so successfully differentiated. Although the model obtained via the canonical discriminant analysis showed a very high discrimination rate (95%), the cross-validation showed that the model was not stable. The correct classification rate of only 45% shows that the original model is not robust enough and is based only on small differences between repetitions. The model becomes more unstable as soon as 1-1 measurement data is omitted from the modelling process, and this one omitted sample has to be predicted by the model. This finding suggests that changing the diet of the animals involved in the study did not cause significantly enough differences in their milk that they could be detected by the electronic nose.

## 5. Conclusions

A low-level of (daily 5 g/head) marine algae supplementation in the goats’ diet did not greatly affect the yield and composition of their milk. In contrast, the results of this study demonstrate that the MA-enriched diet significantly influenced the cheese, whey fat, and protein composition, as well as the recovery of fat and protein in the curd. Moreover, MA supplementation in particular had higher rumenic acid and docosahexaenoic fatty acid content and a lower n-6/n-3 rate than the milk and cheese from the control group.

It can be concluded that changing the diet of the studied animals did not cause such significant differences in their milk that these differences could be detected by the electronic nose.

Our results demonstrate that daily supplementation of 5 g of marine algae has a substantial effect on cheese-processing parameters. Additionally, low-level marine algae supplementation is suitable for the production of healthier goat milks and cheeses (higher concentrations of DHA and rumenic acid) without negatively affecting their sensory profiles.

## Figures and Tables

**Figure 1 animals-13-02152-f001:**
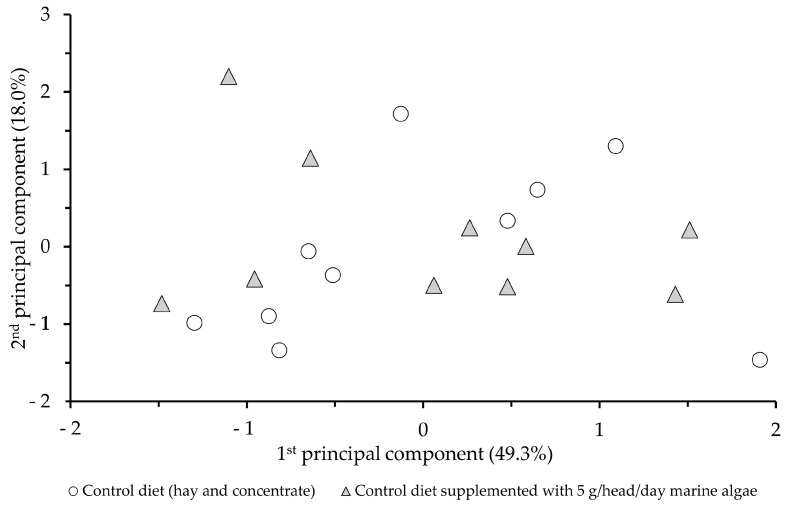
Effect of marine algae inclusion on flavour profile of goat milk, score plot of principal component analysis of electronic nose data.

**Table 1 animals-13-02152-t001:** Chemical composition and fatty acid (FA) profile of fed forage.

Items		Forage		Diet	
	Pasture	Concentrate	Marine Algae	Control ^1^	Marine Algae ^2^
Daily intake g/day	5210	600	5	5810	5815
Daily dry matter (DM) intake g/day	1589	540	4.645	2129	2133
Chemical composition					
DM, g/kg forage	305	900	929	366.36	366.84
Crude protein, g/kg DM	169	170	148	169.18	169.13
Crude fat, g/kg DM	37	26	482	33.84	34.80
Crude fibre, g/kg DM	239	65	23	194.56	194.18
Crude ash, g/kg DM	69	58	38	66.58	66.52
Main FA, g/100g of fatty acids					
C12:0	0.33	nd	0.50	0.27	0.27
C14:0	0.66	0.13	7.28	0.56	0.76
C16:0	11.26	13.05	59.10	11.61	13.06
C18:0	1.97	1.74	2.04	1.93	1.93
C18:1n-9	21.19	26.23	0.71	22.17	21.53
C18:2n-6	18.16	56.84	0.13	25.70	24.99
C18:3n-3	38.95	1.82	0.10	31.71	30.68
C22:6n-3 (DHA)	nd	nd	30.10	nd	0.91
Daily DHA intake, mg/g DM	nd	nd	145.08	nd	0.316
Daily DHA intake, mg/day	nd	nd	673.91	nd	673.91

^1^ Control—control diet (grass and concentrate); ^2^ marine algae—control diet supplemented with 5 g/head/day marine algae; DHA—docosahexaenoic acid; nd—not detected.

**Table 2 animals-13-02152-t002:** Description of the cheese processing.

Processing steps	Description
Heating of raw milk	Temperature: up to 65 °C for 10 min
Cooling	down to 36 °C
Addition of commercial rennet to milk	Hannilase powder commercial rennet (before adding: coagulant diluted by 20 °C water) and gentle stirring for 3 min.
Coagulation of milk	up to 40 min
Post-heating treatment	Temperature: up to 42 °C, duration 5 min
Scooping and draining of coagulated milk	size: 1 cm
Formation of cheeses blocks	Approx. 300 g of each
Whey removal	self-pressing (gravity method), duration: 24 h
Salt bath of cheese blocks	Salt concentration: 18%, salt temperature: 13 °C, duration: 6 h

**Table 3 animals-13-02152-t003:** Effect of marine algae inclusion on daily milk yield and chemical composition of goat milk.

Traits	Pre-Treatment		SEM	*p*-Value	Diet		SEM	*p*-Value
	C	MA			C	MA		
DMY, kg	1.33	1.31	0.017	0.637	1.23	1.21	0.030	0.712
Fat, %	3.60	3.68	0.029	0.191	3.51	3.59	0.041	0.334
Protein, %	3.25	3.29	0.024	0.421	3.24	3.22	0.031	0.777
Lactose, %	4.45	4.43	0.017	0.589	4.47	4.41	0.023	0.102
Total solids, %	12.00	12.10	0.032	0.146	11.92	11.93	0.060	0.270

C—control diet (hay and concentrate), MA—control diet supplemented with 5 g/head/day marine algae, DMY—daily milk yield, kg, calculated as follows: daily bulk milk yield/15 (No of animals).

**Table 4 animals-13-02152-t004:** Effect of marine algae inclusion on cheese yield, chemical composition and nutrient recovery rate of goat cheese.

Traits	Diet		SEM	*p*-Value
	C	MA		
Actual cheese yield, kg/100 kg of milk	12.74	12.03	0.682	0.982
Moisture-adjusted cheese yield *, kg/100 kg of milk	14.32	13.00	0.258	0.009
MFFB, % ^1^	69.42	72.28	0.670	0.033
Fat, %	22.14	21.65	0.420	0.582
Protein, %	20.08	17.83	0.450	0.010
Fat in TS, %	48.37	49.23	1.176	0.731
Total solids, %	46.10	44.10	0.707	0.181
Nutrient recovery rate				
- fat, %	79.61	72.42	1.521	0.017
- protein, %	77.73	66.55	2.221	0.010

C—control diet (hay and concentrate); MA—control diet supplemented with 5 g/head/day marine algae; ^1^ MFBB—moisture content on a fat-free basis; TS—total solids; * adjusted to 44% moisture content.

**Table 5 animals-13-02152-t005:** Effect of marine algae inclusion on whey composition.

Traits	Diet		SEM	*p*-Value
	C	MA		
Fat, %	0.60	1.04	0.056	<0.001
Protein, %	0.85	1.20	0.017	<0.001
Lactose, %	4.38	4.34	0.020	0.308
Total solids, %	6.37	7.15	0.104	0.389
Fat in TS, %	11.97	14.30	0.473	0.012
Protein in TS, %	13.01	16.87	0.490	<0.001
Lactose in TS, %	66.92	60.96	0.875	<0.001

C—control diet (hay and concentrate); MA—control diet supplemented with 5 g/head/day marine algae; TS—total solids.

**Table 6 animals-13-02152-t006:** Effect of marine algae inclusion on fatty acid profile of goat milk and cheese (g/100 g of fatty acids).

Fatty Acids	Milk				Cheese			
	C	MA	SEM	*p*-Value	C	MA	SEM	*p*-Value
C4:0	1.76	1.72	0.071	0.647	2.50	2.00	0.111	0.320
C6:0	1.31	1.38	0.121	0.585	2.99	2.92	0.126	0.799
C8:0	1.61	1.75	0.163	0.388	2.56	2.97	0.103	0.059
C10:0	4.88	6.14	0.560	0.046	5.11	7.42	0.327	0.002
C12:0	1.87	2.53	0.138	0.000	2.46	3.42	0.088	0.000
C14:0	6.53	8.77	0.351	0.000	7.62	9.97	0.182	0.000
C14:1	0.04	0.09	0.005	0.000	0.05	0.11	0.004	0.000
C16:0	24.54	30.06	0.555	0.000	24.13	28.36	0.246	0.000
C16:1	0.57	0.63	0.029	0.048	0.55	0.58	0.014	0.255
C18:0	19.79	14.92	0.496	0.000	16.44	11.94	0.339	0.000
C18:1n-9	28.94	24.29	1.321	0.002	27.42	22.63	0.728	0.004
C18:1t11	2.31	2.22	0.072	0.224	2.04	1.98	0.035	0.409
rumenic acid	0.62	0.88	0.024	0.000	0.57	0.77	0.015	0.000
C18:2n-6	2.29	1.72	0.119	0.000	1.99	1.56	0.037	0.000
C18:3n-3	0.96	0.84	0.029	0.002	0.90	0.78	0.016	0.001
C20:3n-6	0.02	0.02	0.002	0.382	0.02	0.02	0.001	0.999
C20:4n-6	0.17	0.18	0.006	0.751	0.15	0.15	0.004	0.681
C20:5n-3 (EPA)	0.08	0.08	0.003	0.377	0.07	0.07	0.002	0.164
C22:5n-3	0.20	0.17	0.007	0.001	0.16	0.14	0.005	0.164
C22:6n-3 (DHA)	0.06	0.25	0.015	0.000	0.05	0.24	0.005	0.000
odd FA	1.81	1.80	0.048	0.894	1.77	1.75	0.015	0.523
SFA	64.69	69.60	1.358	0.002	68.00	73.19	0.785	0.004
MUFA	31.90	27.25	1.304	0.002	30.10	25.34	0.746	0.004
PUFA	4.41	4.14	0.138	0.073	3.90	3.74	0.062	0.192
n-6	2.49	1.92	0.119	0.000	2.16	1.73	0.039	0.000
n-3	1.30	1.35	0.039	0.293	1.18	1.23	0.021	0.112
n-6/n-3 ratio	1.92	1.43	0.103	0.000	1.84	1.41	0.025	0.000
AI	1.47	2.17	0.096	0.000	1.84	2.70	0.130	0.000

C—control group; MA—marine algae group; SCFA—short-chain fatty acids; SFA—saturated fatty acids; MUFA—monounsaturated fatty acids; PUFA—polyunsaturated fatty acids, odd FA—odd-chain fatty acids; AI—atherogenic index.

**Table 7 animals-13-02152-t007:** Calculated docosahexaenoic acid (DHA) conversion efficiency from marine-algae-enriched diet in milk.

Days ^1^	Daily DHA, intake, mg/Day ^2^	Average Milk Fat Content, g/100 g Milk	Average Milk DHA Content, g/100 g FA	Average Milk DHA Content, mg/100 g Milk	Milk Production, kg	DHA in Milk Yield, mg/Day	DHA Efficiency Ratio, %
Adaptation period, 42 d ^3^							
7	673.91	3.68	0.08	2.94	1.23	36.21	5.37
14	673.91	3.71	0.10	3.71	1.22	45.26	6.72
21	673.91	3.75	0.13	4.88	1.21	58.99	8.75
28	673.91	3.78	0.16	6.05	1.23	74.39	11.04
35	673.91	3.81	0.19	7.24	1.22	88.32	13.10
42	673.91	3.85	0.26	10.01	1.22	122.12	18.12
Experimental period 10 d ^4^							
1–10	673.91	3.88	0.25	9.70	1.21	117.37	17.42

^1^ Control diet supplemented with 5 g/head/day marine algae; ^2^ daily DHA intake based on Table 1, 145.08 mg DHA/g DM marine algae; ^3^ calculated weekly collected bulk milk sample from MA group; ^4^ mean value of bulk milk samples were taken each day during experimental period (1–10 days) from MA group.

**Table 8 animals-13-02152-t008:** Effect of marine algae inclusion on flavor profile of goat milk; confusion matrix of canonical discriminant analysis of electronic nose data.

	Original Group Membership	Predicted Group Membership	
		C	MA
Original *	C	100.0	0.0
	MA	10.0	90.0
Cross-validated **	C	40.0	60.0
	MA	50.0	50.0

C—control diet (hay and concentrate); MA—control diet supplemented with 5 g/head/day marine algae; * 95.0% of original grouped cases correctly classified; ** 45.0% of cross-validated grouped cases correctly classified.

## Data Availability

Data are available from the corresponding author upon request.

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
