# Peer review of "Changes in Chemical Composition and Fatty Acid Profile of Milk and Cheese and Sensory Profile of Milk via Supplementation of Goats’ Diet with Marine Algae"

_animals, 2023, doi:10.3390/ani13132152_

Round 1

Reviewer 1 Report

Changes in Chemical Composition and Fatty Acid Profile of Milk and Cheese and Sensory profile of Milk by Marine Algae Supplementation
L 2-4: Changes in Chemical Composition and Fatty Acid Profile of Milk and Cheese and Sensory profile of Milk by supplementation the goats' diet with marine algae.
L24-26: Please rephrase the sentence to indicate the species of animal on which the study was carried out (cows, goats, ewes .........).
L44: docosahexaenoic acid (DHA) and the rumenic acid (CLA c9, t11)
L46-48: Please rephrase the sentence (milk obtained from goats fed daily with 5g MA supplements has a fatty acid profile more favorable to human health without any negative effect on the milk odour components).
L73-74: Please check this data (differences between adults and children are very large and unjustified - in my opinion).

The manuscript titled - Changes in Chemical Composition and Fatty Acid Profile of Milk and Cheese and Sensory profile of Milk by Marine Algae Supplementation - addresses an interesting, topical, well-documented research topic and appears to be well-written. However, it presents some shortcomings that put me in difficulty:

- in the Abstract it is mentioned that the goats were fed with alfalfa hay (1500 g/head/day) (see L 33) and in Materials and Methods it is detailed that the goats consumed grass on the pasture throughout the morning and evening milking ( see L 111-112, L 118) (a big problem).

- it does not result if the two groups of goats grazed together or each group benefited from a separate grazing paddock.

- the method by which weed consumption was determined is inadequate (empirical).

- in table 1 - the data is not correctly presented (e.g. Chemical composition - dry matter, g/kg forage = is it pasture, concentrates, weight average? I think it would be correct to present chemical composition separately for pasture and separately for concentrates - to be in agreement with the way of presentation of the intake. A similar situation is also in the case of Main FA, g/100g of fatty acids. In addition for DHA - to group C is superscript 4 ????. The grass and concentrates contain DHA - more at the bottom of table 1 is passed DHA daily intake: 673.91 mg - for which group of goats???).

I did not continue the review to the end of the manuscript because the conditions experienced are not clear to me - although I am convinced that hard work has been done.

I feel compelled to propose rejection of the manuscript, but if the other reviewers have another proposal I will agree with it.

Author Response

Reviewer 1

Dear Reviewer,

Thank you for your review, it allows to improve our manuscript.

All our corrections are marked in the Text!

You can see our responses to your questions and comments:

L 2-4: Changes in Chemical Composition and Fatty Acid Profile of Milk and Cheese and Sensory profile of Milk by supplementation the goats' diet with marine algae.

  • Thank you for your comment, and thank you for your recommended title, we changed it!

L24-26: Please rephrase the sentence to indicate the species of animal on which the study was carried out (cows, goats, ewes .........).

  • Thank you for your comment, we added the recommended information in the text!

L44: docosahexaenoic acid (DHA) and the rumenic acid (CLA c9, t11)

  • Thank you for your comment, we added these abbreviations in the text!

L46-48: Please rephrase the sentence (milk obtained from goats fed daily with 5g MA supplements has a fatty acid profile more favourable to human health without any negative effect on the milk odour components).

  • Thank you for your comment, and thank you for your recommended sentence, we corrected it!

L73-74: Please check this data (differences between adults and children are very large and unjustified - in my opinion).

  • Thank you for your remark, we checked this data, we corrected it!

- in the Abstract it is mentioned that the goats were fed with alfalfa hay (1500 g/head/day) (see L 33) and in Materials and Methods it is detailed that the goats consumed grass on the pasture throughout the morning and evening milking ( see L 111-112, L 118) (a big problem).

  • Thank you for your comment, unfortunately, it was a mistake in the abstract, all animals were kept on the pasture (without hay diet), we corrected it!

- it does not result if the two groups of goats grazed together or each group benefited from a separate grazing paddock.

  • Thank you for your comment, all goats were grazed together, the difference between diets was in the ingredient of concentrate (with or without marine algae supplementation), we corrected it!

- the method by which weed consumption was determined is inadequate (empirical).

  • Thank you for your comment, concentrate and marine algae was dosed (600 and 5 g) without any refusals. The grass yield was determined by hand clipping pre- and post-grazing period, we specified and corrected the green yield determination method (lines 138-142)!

- in table 1 - the data is not correctly presented (e.g. Chemical composition - dry matter, g/kg forage = is it pasture, concentrates, weight average? I think it would be correct to present chemical composition separately for pasture and separately for concentrates - to be in agreement with the way of presentation of the intake. A similar situation is also in the case of Main FA, g/100g of fatty acids. In addition for DHA - to group C is superscript 4 ????. The grass and concentrates contain DHA - more at the bottom of table 1 is passed DHA daily intake: 673.91 mg - for which group of goats???). In addition for DHA - to group C is superscript 4 ????. The grass and concentrates contain DHA - more at the bottom of table 1 is passed DHA daily intake: 673.91 mg - for which group of goats???).

  • Thank you for your comment, we gave the chemical and fatty acid composition and forage intakes in Table 1 separately by forages, moreover by diets too. The superscript (#4) is miss typing, DHA content was added by type of forages, nevertheless, in the concentrate and grass samples were not detected DHA, only in marine algae. The calculated daily DHA intake was only in the experimental diet (which contains marine algae) and we added this information in Table 1.

Reviewer 2 Report

In general, all parts of the article are written in understandable language and the transitions between sections are well organized. I would like to congratulate the authors for the good quality of the article, the literature reported used to write the paper, and for the clear and appropriate structure. The manuscript is well written, presented and discussed, and understandable to a specialist readership. The organization and the structure of the article are satisfactory and in agreement with the journal instructions for authors. The subject is adequate with the overall journal scope.

Here are some suggestions I made to improve the article;

1.    In order to further improve the quality of the paper, an overall check of the English language is recommended.

2.    Did you take into account the order of lactation in the study? This is not specified.

3.    In line 414 References are not numbered, please check.

4.    In line, 425 "cc 40%" is not clear what you mean, be more specific.

5.    In line 414; "Białek et al., 2018b" It seems that you did not use other articles by these authors published in the same year in your study. If not, you can delete "b".

In order to further improve the quality of the paper, an overall check of the English language is recommended.

Author Response

Reviewer 2

Dear Reviewer,

Thank you for your review, it allows to improve our manuscript.

All our corrections are marked in the Text!

You can see our responses to your questions and comments:

  1. In order to further improve the quality of the paper, an overall check of the English language is recommended.
  • Thank you for your recommendation, we improved the quality of the English language!

  1. Did you take into account the order of lactation in the study? This is not specified.
  • Thank you for your comment, we indicated the parity of the experimental animals in the Materials and Methods section!

  1. In line 414 References are not numbered, please check.
  • Thank you for your comment, we corrected them!

  1. In line, 425 "cc 40%" is not clear what you mean, be more specific.
  • Thank you for your question, we rewrite and specified this sentence!

  1. In line 414; "Białek et al., 2018b" It seems that you did not use other articles by these authors published in the same year in your study. If not, you can delete "b".
  • Thank you for your recommendation, we deleted the “b” mark!

Reviewer 3 Report

Comment to Authors

As reported by the authors this article evaluates the effect of marina algae supplementation in the diets   of goats on the chemical composition and fatty acid profile of milk and cheese 

The paper is interesting, therefore are needed minor revisions.

Some points must be attended to before publication:

 Line 33 The experimental diet as reported in Table 1 is different. Please improve the sentence.

Line 118 Please indicate how marine algae is provided in the diet

Table 6 please change c11 C18:1n-9  to  C18:1n-9 and t11 C18:1n-7 to C18:1t11

Line 296

The authors report that DHA intake is 673 mg for the whole duration of the experiment. The duration of the experiment is 52 days of which 42 days are adaptation and 10 days of the experimental period, thus based on this, the table 7 should be reviewed and corrected as regards the daily intake of DHA and day.

 Line 314 please delete that distinct groups could be 314 obtained by principal component analysis.

 Line 396 please delete and more

Line 414 please delete (Toral et al., 2017; Białek et al., 2018b)

 Line 425 please delete cc

 Best regards

Author Response

Reviewer 3

Dear Reviewer,

Thank you for your review, it allows to improve our manuscript.

All our corrections are marked in the Text!

You can see our responses to your questions and comments:

Line 33 The experimental diet as reported in Table 1 is different. Please improve the sentence.

  • Thank you for your comment, we corrected this sentence!

Line 118 Please indicate how marine algae is provided in the diet

  • Thank you for your question, we rewrite and add more information about marine algae providing (lines 124-125)!

Table 6 please change c11 C18:1n-9  to  C18:1n-9 and t11 C18:1n-7 to C18:1t11

  • Thank you for your comment, we changed them!

Line 296

The authors report that DHA intake is 673 mg for the whole duration of the experiment. The duration of the experiment is 52 days of which 42 days are adaptation and 10 days of the experimental period, thus based on this, the table 7 should be reviewed and corrected as regards the daily intake of DHA and day.

  • Thank you for your comment, we corrected this Table, we indicated the two terms separately according to adaptation and experimental periods!

Line 314 please delete that distinct groups could be 314 obtained by principal component analysis.

  • Thank you for your comment, we deleted the recommended phrase!

Line 396 please delete and more

  • Thank you for your comment, we deleted the “and more”!

Line 414 please delete (Toral et al., 2017; Białek et al., 2018b)

  • Thank you for your comment, we corrected them!

Line 425 please delete cc

  • Thank you for your comment, we deleted the “cc”!

Round 2

Reviewer 1 Report

The manuscript has been sufficiently improved to warrant publication in Animals.